# Measuring the Demand Connectedness among China’s Regional Carbon Markets

**DOI:** 10.3390/ijerph192114053

**Published:** 2022-10-28

**Authors:** Li-Yang Guo, Chao Feng

**Affiliations:** School of Economics and Business Administration, Chongqing University, Chongqing 400030, China

**Keywords:** emission-trading market, trading volume, demand for emission allowances, connectedness, frequency decomposition

## Abstract

After years of emission trading in segmented pilots, China operates a unified market in the power system and plans to involve more industries in the coming future. The aim of this study is to detect the commonalities of transaction behaviors across China’s regional carbon pilots, so as to provide an empirical basis for a future multi-sectoral expansion of national trading. Based on a dataset of daily trading volume in seven regional markets during 2014–2021, the empirical results from connectedness measures show that the total demand connectedness ranges from 10% to 24%, indicating the existence of interactions among China’s regional markets. This not-so-wide range of fluctuation usually shows a trend of rising first and then falling within each year, during which the upward trend is basically related to the accounting, verification and compliance of allowances. After these time nodes, the total connectedness declines. In addition, the directional connectedness could help clarify the specific roles that regional markets play in the variations of total demand connectedness when facing the shocks of these time nodes. Meanwhile, the frequency decomposition reveals that a longer-term component of more than 10 days dominates the connectedness. Based on these findings, some policy implications are provided alongside.

## 1. Introduction

As a contracting party of the Paris Agreement, and as one of the major emitters in the world [1], China is actively assembling the capabilities of the entire industry, striving to achieve carbon dioxide neutrality by 2060 and net-zero emission of greenhouse gases as soon as possible, which means that it is necessary for China to reduce carbon dioxide emissions by a large proportion [2]. In order to further promote China to move towards a low-carbon economy, the government has proposed many green policies to guide economic transformation [3]. One of the policy tools is the operation of the carbon-emission-trading pilots, which includes some major regions of China, and covers several carbon-intensive industries. After nearly 10 years of the regional pilot phase, China officially started a national emission-trading system for the power industry in 2021, forming a situation in which the regional markets and the national system are operated in parallel.

With the trial implementation of China’s dual-track carbon emission market, more and more high-energy-consuming companies have been covered in this process. While dealing with carbon-emission-related risks [4], the financial nature of allowances can also help traders obtain investment income. Regardless of the purpose of the transaction, the participating companies have worked together to promote the maturity and construction of the markets [5]. Although China plans a multi-sectoral expansion of national trading, heterogeneity exists in the current market conditions across the pilot regions. With resource constraints, the industrial structure partly dominates the structure of energy consumption, which mainly determines the overall constraint capacity of the total amount of local emission allowances, and in turn affects the trading activity and trading behavior of the market. Meanwhile, the large differences in the environmental and climatic conditions within China’s territory also lead to heterogeneous electricity-consumption habits, which in turn affect electricity-related energy consumption, and further determine CO_2_ emissions and allowance-trading behaviors. In addition, the speed of technological progress in each region affects the surplus of allowances in production, which in turn forms a differentiated behavior of allowance trading across regions. So, these inevitably raise the questions of whether there are interactions among transaction behaviors across China’s carbon-trading system, how much commonality the trading behavior has across the regions, and how long this commonality can last.

Undoubtedly, the answers to these questions help to clarify the impact of changes in allowance demand on the entire emission-trading system and to determine the common demand behavior of the markets in the face of exogenous shocks. In order to address them, this study selects seven relatively matured emission-trading pilots. The national market (power industry) is not considered, which has few trading days compared to other markets and is not conducive to providing a panoramic analysis. The daily trading volumes are regarded as the total daily demands in the carbon markets, which will be the material of the connectedness measures. The rolling-window estimation is applied to the connectedness to generate a basis for the dynamic analysis in the time domain. Finally, the frequency decomposition helps to split the dynamic connectedness into different time bands, which can be individually defined as short-, medium-, and long-term frequency. Using such an empirical design, while the dynamic characteristics and frequency distribution of the connectedness across carbon markets are clarified, we also analyze the drivers of connectedness over time based on the reality of emission trading and economic activity. This facilitates an understanding of the commonalities between regions in the carbon market and helps to provide an empirical basis for a future multi-sectoral expansion of national trading.

This study mainly contributes in two important ways. Firstly, this study adds to the literature by identifying the common cross-market behaviors from trading volumes, which can more directly reflect the daily demands for allowances in the carbon markets, whether for speculation or actual production. Price movements, as the research material of carbon markets, reflect the process of trading. But their portrayal of the decisive behavior of emission trading does not fit our conception of this research topic. Therefore, the volume is relatively more conducive to playing a role in the connectedness of the demand behavior across the markets. Secondly, the empirical evidence from this study helps to inform the construction of carbon markets. The frequency connectedness measures help to clarify the degree of interactions across the regional carbon markets in both time and frequency domains, and their time-varying characteristics and driving force are also identified. Based on this, environmental authorities, participating companies and investors can grasp the potential common changes of demand in the emission-trading system a priori, and be provided with an empirical basis for operating a nationwide unified trading system in a larger industry in the future.

The remainder of this study consist of four sections. Section 2 briefly reviews the related literature. Section 3 introduces the methodology and data. Section 4 reports the empirical results. Section 5 gives the conclusions and implications.

## 2. Literature Review

### 2.1. Benefit and Risk of Emission-Trading Policy

At present, the effect of this policy tool in China has been demonstrated to a certain extent. The CO_2_ emissions of the industrial sector have been reduced [6,7], which also co-benefits the environment [8], as well as the obvious reduction in regional CO_2_ intensity [9]. Such effects are also found with EU emission-trading system [10,11]. Meanwhile, since the long-term emissions of enterprises may require greater costs, and in order to adapt to the green transformation of the entire society, carbon emission trading can also be said to promote green innovation and low-carbon technological progress [12,13,14,15]. In addition, participating in emission trading could help promote the development of companies [16,17]. Obviously, the implementation of carbon emission trading could exhibit a positive effect on the environment and enterprises in the long run.

However, the carbon market has similar trading characteristics to the secondary financial market. Transactions of emission trading may be related to other markets, and then form a systematic cross-influence. The relationships between them are hence being enthusiastically discussed. While being positively correlated with the stock returns of market participants [18], the prices of emission allowances totally interact with energy sectors, such as fossil fuels [19]. These performances are similar to those of the mature EU market, such as the relationships between the emission-trading market and energy derivatives or carbon-intensive companies [20,21,22], as well as the information connection of emission trading and non-energy commodities [23]. However, some opposite views state that there is no significant cross-effect between emission trading and stock returns in the EU market [24,25].

### 2.2. Connectedness of Emission-Trading Markets

With the consideration of a future multi-sectoral expansion of national emission trading, institutions from various industries will also be involved. Then, does the emission trading under the segmented market form a certain degree of commonality among regions? Figuring this out is clearly crucial for building a national market. Specifically, the prices of China’s major emission-trading markets, including Guangdong, Shenzhen and Hubei, are confirmed to show the co-movement dynamically, implying further opportunities for the establishment of a national market [26]. Emission trading also provides a channel for risk transmission, although this is locally found between the Guangdong market and the Shenzhen market [27]. However, in terms of tail dependence, the Hubei market is inseparable from all the others [28]. Although spillovers of two types of prices are found in more market scales—seven major pilots in total—the effects they have are relatively small [29,30], indicating that there is still a certain difference among the markets. In addition to pricing gaps, more differences may come from the extent to which market prices respond to information flows [31,32], which may result from the low awareness of participants, prohibition of the cross-regional flow of allowances, and differentiated market effectiveness [33,34,35].

All of the above types of connectedness, whether partially or globally, are basically generated from multiple-variable systems, such as Copula and the VAR-based approach. Guo and Feng (2021) and Xiao et al., (2022) applied this kind of VAR-based approach to quantitatively examine the contribution of each market to the connectedness [29,30]. This connectedness framework was proposed by Diebold and Yilmaz [36,37], which calculates the portions of variance variations in a certain variable due to the shock arising from elsewhere, and is widely used in various kinds of financial markets [38,39,40]. Baruník and Křehlík (2018) further developed this framework with spectrum causality [41], allowing for the connectedness analysis in frequency domains [42,43,44,45]. Following Guo and Feng (2021) [29], this study also considered the connectedness in frequency domains, which aimed to measure the interactions in China’s emission trading for different time bands.

### 2.3. Review Summary

Whether the interconnection of the carbon emission system itself or its connection with the financial markets, they both remind companies or future speculators that they may have some external effects when trading allowances in the markets. Since different traders can bear or expect different carbon allowance prices, they will trade at different prices according to their own capabilities. Such capacity differences will lead to different emission allowance demands in the market within a certain period of time. Therefore, unlike the previous studies, this study will start with the realized transactions, taking the transaction volume of emission allowances as the main research object and exploring whether there is a certain commonality between the segmented markets in the formation of demand for allowances, especially at the time nodes during which the carbon market experiences special shocks. Accordingly, this study could help to clarify the common transaction behaviors, hence providing an empirical basis for a future multi-sectoral expansion of national trading.

## 3. Methodology and Data

### 3.1. Frequency Connectedness Measure

To analyze the demand interactions of China’s emission-trading markets, both in time and frequency domains, it is necessary to clarify the influences of a certain market on the others’ demands, as well as the demand variation caused by others. This kind of trading behavior caused by cross-market shock flows can be clearly expressed by a “connectedness” measure of Diebold and Yilmaz (2012, 2015) [36,37]. They proposed and developed a mature analysis framework, which measures the variance variation of a certain variable due to the shocks arising from elsewhere. The most obvious feature of this method is that it overcomes the dependence on variable ordering caused by the Cholesky orthogonalization. By such a foundation, Barunik and Krehlik (2018) decomposed the connectedness into different time domains [41], allowing the frequency analysis of connectedness and the definitude of its most contributed period of time. The entire methodology of frequency connectedness was originally designed for stationary financial time series, such as return and volatility. Here, this study applies it to stationary series of trading volume, which stand for the demand in emission-trading markets. Considering an *N*-dimensional covariance stationary process $x_{t}=(x_{1,t},x_{2,t},\cdots x_{N,t}{)}^{\prime }$ formed as a VAR system:(1)$$B(L)x_{t}=\varepsilon_{t},$$
where $B(L)=\left[I-B_{1}L-\cdots -B_{p}L^{p}\right]$ refers to the lag polynomial matrix with order p, each $x_{j,t}$ of $(x_{1,t},x_{2,t},\cdots x_{N,t}{)}^{\prime }$ stand for the final demands in the jth emission-trading markets at time t, respectively. Easily, a polynomial $\Gamma (L)={\left[B(L)\right]}^{-1}$ can always be found to transform the VAR(p) into its VMA(*∞*) form $x_{t}=\Gamma (L)\varepsilon_{t}$, and the matrix Γ(L) with infinite lags can express the impulse response at h=1,…,H horizons, which can be labeled as $\Gamma_{h}$ and is the fundamental of variance decomposition.

Based on the generalized forecast error variance decomposition [46,47], which is invariant to variable ordering, the contribution arising from the shocks in the kth market to the forecast error variance of the demand in the jth market can be expressed as:(2)$$d_{j,k}(H)=\frac{\sigma_{kk}^{-1}\sum_{h=0}^{H-1}{\left({\left(\Gamma_{h}\Sigma \right)}_{j,k}\right)}^{2}}{\sum_{h=0}^{H-1}{\left(\Gamma_{h}\Sigma {\Gamma^{\prime }}_{h}\right)}_{j,j}},$$
where Σ is the covariance matrix with the kth diagonal element being $\sigma_{kk}$, and *H* refers to the error forecast horizons of the demand in the kth market. Due to the row sum of d(H) not being equal to one, each entry can be normalized as:(3)$${\tilde{d}}_{j,k}(H)=\frac{{{}_{}}_{}{}_{}d_{j,k}(H)}{\sum_{k=1}^{N}d_{j,k}(H)},$$
where $\sum_{k=1}^{N}{\tilde{d}}_{j,k}(H)=1$ and $\sum_{j,k=1}^{N}{\tilde{d}}_{j,k}(H)=N$. Therefore, the total connectedness can be calculated as:(4)$$C(H)=\frac{\sum_{j\neq k}^{}{\tilde{d}}_{j,k}(H)}{N}\cdot 100$$

This total connectedness measures the contributions arising from the shocks across all the markets to the demand uncertainty of the entire emission-trading system. Thereby, the directional connectedness from other markets in the emission-trading system to the demand variation in the jth market (“from connectedness”) can be defined as:(5)$$C_{j\leftarrow \cdot }(H)=\frac{\sum_{\begin{array}{l} k=1\\ j\neq k \end{array}}^{N}{\tilde{d}}_{j,k}(H)}{N}\cdot 100,$$
and the connectedness from the jth market to all the others (“to connectedness”) is:(6)$$C_{\cdot \leftarrow j}(H)=\frac{\sum_{\begin{array}{l} k=1\\ j\neq k \end{array}}^{N}{\tilde{d}}_{k,j}(H)}{N}\cdot 100.$$

Next, the key to construct frequency connectedness is to understand how the forecast error variance of the demands in each market is distributed over the frequency ω. To deal with this, the impulse response $\Gamma_{h}$ should be expressed as its spectral representation through a Fourier transform, which is $\Gamma (e^{-i\omega })=\sum_{h}e^{-i\omega h}\Gamma_{h}$ with $i=\sqrt{-1}$. Then, the spectrum density, showing such distribution of variance, can be described as $S(\omega )=\Gamma (e^{-i\omega })\Sigma \Gamma^{\prime }(e^{+i\omega })$. Similar to the implication of the variance decomposition, the spectrum share of the demand in the jth market due to shocks arising from the kth market at frequency ω can be calculated as:(7)$$\xi_{j,k}(\omega )=\frac{\sigma_{kk}^{-1}{\left[{\left(\Gamma (e^{-i\omega })\Sigma \right)}_{j,k}\right]}^{2}}{{\left(\Gamma (e^{-i\omega })\Sigma \Gamma^{\prime }(e^{+i\omega })\right)}_{j,j}}.$$

In order to perform the variance decomposition with frequencies, $\xi_{j,k}(\omega )$ should be given weights according to the portions accounted for by the jth market’s variance, which is defined as:(8)$$W_{j}(\omega )=\frac{{\left(\Gamma (e^{-i\omega })\Sigma \Gamma^{\prime }(e^{+i\omega })\right)}_{j,j}}{\frac{1}{2\pi }\int_{-\pi }^{\pi }{\left(\Gamma (e^{-i\lambda })\Sigma \Gamma^{\prime }(e^{+i\lambda })\right)}_{j,j}d\lambda },$$
and setting the sums of all frequencies to 2π. Actually, the analysis of frequency connectedness may have to be concentrated on a certain time period, hence the variance decomposition with a specific frequency band m can be expressed as:(9)$$d_{j,k}(m)=\frac{1}{2\pi }\int_{m}W_{j}(\omega )\xi_{j,k}(\omega )d\omega ,$$
where m=(a,b):a,b∈(−π,π),a<b is the specific frequency band to be set in the subsequent analysis. Similar to Equation (4), each entry of this variance decomposition can be normalized as:(10)$${\tilde{d}}_{j,k}(m)=\frac{{{}_{}}_{}{{}_{}}_{}d_{j,k}(m)}{\sum_{k=1}^{N}d_{j,k}(\infty )}.$$

Note d(∞) indicates the variance decomposition on (−π,π), which covers all the frequencies. The total frequency connectedness on a specific frequency band m can be calculated by
(11)$$C(m)=(\frac{\sum \tilde{d}(m)}{\sum \tilde{d}(\infty )}-\frac{Tr\{\tilde{d}(m)\}}{\sum \tilde{d}(\infty )})\cdot 100,$$
where $Tr\{\tilde{d}(m)\}$ indicates the trace of $\tilde{d}(m)$. The computational procedure is referred to the open-source R package “frequencyConnectedness” (Tomas Krehlik (2020). frequencyConnectedness: Spectral Decomposition of Connectedness Measures. R package version 0.2.3 (https://CRAN.R-project.org/package=frequencyConnectedness, accessed on 1 February 2022.)).

### 3.2. Data and Preliminary Analysis

Under the current dual-track emission-trading scheme, there are 10 markets for emission trading, consisting of 9 regional pilot markets and a national (power industry) market. In order to perform a connectedness analysis of the demand of emission allowances, this study selects seven long-established and relatively matured markets, which are Chongqing (CQEA), Shenzhen (SZEA), Hubei (HBEA), Beijing (BJEA), Shanghai (SHEA), Tianjin (TJEA) and Guangdong (GDEA). Sichuan market is excluded for its trading of Chinese Certified Emission Reduction (CCER), and Fujian and the national markets are excluded for having the shortest trading days. The data sample composed of trading volumes in seven markets covers the time period from 19 June 2014 to 31 March 2021. The demand for emission allowance in a certain market can be defined as:(12)$$Q_{j,t}=ln(vol_{j,t}+1),$$
where $vol_{j,t}$ is the trading volume of the jth market at date t, the expression $\left(vol_{j,t}+1\right)$ is to ensure that logarithm transformations can also be performed when zero trading volume appears. It is necessary to note that the annual allowance in the Shenzhen market before 2020 will be allocated with an annual mark and listed for trading at a differentiated price, such as SZEA-2015. It can only be used to execute the compliance of 2015 and beyond. Since the daily demand for all emission allowance products in Shenzhen is also very different, it is biased to only use the transaction volume of a certain product to represent the market demand of Shenzhen. Therefore, referring to Guo & Feng (2021) [29], this study selects the most active products in each year as a representative, which is mostly traded during that year, thereby forming the time series of the trading volume in Shenzhen. The dynamic development and descriptive statistic of emission allowance demand in seven markets are shown in Figure 1 and Table 1, respectively.

It can be clearly seen from the Table 1 that the minimum demand for emission allowances in every market is zero, which also means that there are days that no transactions are made. The logarithmic volume in each market ranges widely from 0 to 6, while the Shanghai market has the largest standard deviation, and the Hubei market has the smallest one. The *p*-values for the ADF test all indicate the stationarity of the demand series. In addition, it can be inferred from the values of the mean that the Guangdong and Hubei markets have a relatively high demand for carbon allowances, but the demand in the Chongqing and Tianjin markets is relatively small, which may also be inferred from Figure 1. Panels (1) and (6) of the figure show that the Chongqing and Tianjin markets are very inactive for some periods. In addition, the demand in each market shows a sharp increase in the first half of each year, and their time periods coincidentally correspond to the time window of annual verification and compliance matters, which may be the main source of shock to the carbon markets.

## 4. Empirical Analysis

In this section, the empirical results are reported. As the above discussions, to measure how the demand of all the emission-trading markets interact, the connectedness framework is applied to express the uncertainty of the demand in a certain market due to the shocks arising from elsewhere [36,37]. Meanwhile, the frequency analysis of these shocks is also crucial to clarify how their impacts distributed over some specific time periods, which can be estimated through frequency connectedness [41]. At the same time, some parameters need to be preset in the estimation, such as the lags of the VAR process and the forecast horizons of the variance decomposition; different specifications may lead to different estimation results. Therefore, to determine how sensitive the connectedness is to these parameter values, a specific analysis is performed at the end of the empirical analysis.

### 4.1. Connectedness in Time Domain

For a seven-dimensional VAR process corresponding to the demand for emission allowance, four lags are selected according to the information criteria and the forecast horizons are set to be 100 days. The total demand connectedness is estimated under these specifications, the dynamics of which are displayed in Figure 2. A 330-day rolling-window estimation is carried out to fit the time-varying connectedness, with the first value appearing at 20 October 2015. From Figure 2, the total demand connectedness approximately ranges from 10% to 24%, which indicates the scale of the contributions arising from the shocks across all the markets to the demand uncertainty of entire emission-trading system. That is to say, the aggregate degree of the impact on the demand in each market that is affected by the demand in other markets fluctuates between 10% and 24%. The maximum value of this kind of cross-market impact appears in September 2020, and the relatively smallest value appears from January to February 2019. In short, there is a certain degree of interaction behavior across the demand in China’s regional carbon allowance, the degree of which is measured by the variance variation of the demand caused by the shock transmission. Generally, the total demand connectedness exhibits a cyclical nature, which can be roughly summarized as the basic trend of rising first and then falling in fluctuation within each year. From the meaning of total demand connectedness itself, its rising trends mean that the proportions of variance variation of each market in the entire carbon-trading system caused by other markets are increasing, and the same as falling trends.

Therefore, a detailed analysis of demand connectedness is performed to determine the reasons for its trend variation and to clarify the contributions of each market to the entire system, and the comparisons are combined in Figure 3 and Figure 4, which individually show “to connectedness” and “from connectedness”. Specifically, within 2016, the total demand connectedness rises first and then decreases from January to the end of May. During this period, the “to connectedness” of BJEA (Figure 3, panel 4) clearly shows an upward trend as well as large fluctuations in demand (Figure 1, panel 4) until the end of March, which implies the increasing influence that shocks in the demand in BJEA had on the demand uncertainty of other markets. The large fluctuations in BJEA may be due to the announcement of specific emission accounting and the verification need to be done before the end of March. Therefore, the demanders of emission allowances carry out transactions of varying degrees, and the behavior and information are transmitted to other allowance markets, which may mainly be SZEA (Figure 4, panel 2) and SHEA (Figure 4, panel 5). After this and before the end of May, as others have made few contributions to the system, a short downward trend occurs in the “to connectedness” of BJEA (Figure 3, panel 4) as well as a steady decline in SHEA (Figure 3, panel 5), which results in an obvious decline in total demand connectedness. Similarly, after entering the compliance procedure, “to connectedness” of BJEA exhibits an upward trend from the end of May to the end of June and a long-term downward trend thereafter, and the impact from the shocks in BJEA is basically received by SZEA (Figure 4, panel 2) and SHEA (Figure 4, panel 5).

Within 2017, the fluctuations in total demand connectedness are similar to those of 2016, but they experienced another cycle from rising to falling in the second half of 2017; the three obvious peaks in this year may all be related to CQEA (Figure 3, panel 1). From April to mid-June, the demand for CQEA (Figure 1, panel 1) first rises rapidly to a certain level and then forms sharp fluctuations around this area. This may be due to preparation of emission accounting and verification procedures based on the experience of previous years in CQEA, because these notices are officially announced in the second half of the year. At the same time, compliance procedures are being implemented in other markets, so they are also sensitive to abnormal fluctuations in CQEA (Figure 1, panel 1). During this period, the impacts caused by the uncertain shock in CQEA are mainly received by SZEA (Figure 4, panel 2) and BJEA (Figure 4, panel 4), and the “from connectedness” of TJEA (Figure 4, panel 6) and GDEA (Figure 4, panel 7) are strengthened by the comprehensive influence from other markets. Additionally, from August to early September, the demand fluctuations in each market are all relatively obvious, and the increasing level of interactions among them enhance the total demand connectedness, which may arise from BJEA (Figure 3, panel 4), TJEA (Figure 3, panel 6), and GDEA (Figure 3, panel 7), and SZEA (Figure 4, panel 2) and SHEA (Figure 4, panel 5) may receive relatively more of these shocks.

During 2018, an obvious peak appears between April and July. However, before April, the total demand connectedness shows an unstable and volatile trend. During this period, the shocks coming from GDEA (Figure 3, panel 7) are maintained at a relatively high level, and the CQEA compliance of the previous year is not finally completed until May 2018, providing signals for other markets to pay attention to emission verification and compliance, resulting in an upward trend of “to connectedness” in CQEA (Figure 3, panel 1). The main recipients of these shocks are HBEA (Figure 4, panel 3) and BJEA (Figure 4, panel 4). From then, the total demand connectedness begins to rise again until late June, during which the compliance procedure is being carried out. The contribution to the demand uncertainty of the entire system may be mainly dominated by BJEA (Figure 3, panel 4) and GDEA (Figure 3, panel 7), and HBEA (Figure 4, panel 3) may receive relatively more shocks than others. After about June 20th to the end of the year, the degree of cross-market interactions of the emission-trading system shows a steady downward trend, which can be seen from the total demand connectedness and implies that the impact arising from elsewhere on the uncertainty of each market is diminishing.

For 2019, the total demand connectedness experiences a brief rise from February to mid-April, which is the time that most of the markets are accounting and verifying their CO_2_ emissions. The contributors are mainly SZEA (Figure 3, panel 2), BJEA (Figure 3, panel 4) and SHEA (Figure 3, panel 5). However, from Figure 4, it can be seen that the variance variations of SZEA and BJEA are also affected relatively more by other markets, which may imply that the shocks in the demand for emission allowances mainly transmit among these three markets. Then, from the beginning of June, compliance matters are being implemented in all the markets except CQEA, and the demand fluctuates to a relatively larger degree than before, that is, uncertainty increases. The total demand connectedness hence rises again until compliance ends for most markets in mid-August, the uncertainty of demand in HBEA (Figure 4, panel 3) receives relatively more impacts. The downward trend of total connectedness thereafter is easy to understand. In the period after the completion of compliance, no common shocks such as accounting, verification or compliance spread across the market, so similar changes in demand occur.

When it comes into 2020, due to the large-scale impact of the COVID-19 epidemic, the accounting, verification and compliance of carbon emissions are all delayed and last longer than before. It can be clearly seen from Figure 2 that the total connectedness increases in volatility until early September. Without knowing the actual time that local authorities decide to implement compliance while controlling the COVID-19 epidemic, emission traders in every market are always preparing for related work, so the trading of emission allowance is very frequent, resulting in a rising trend of total demand connectedness in volatility. TJEA (Figure 3, panel 6) and GDEA (Figure 3, panel 7) make relatively high contribution to this system before August, and the relatively greater portion of shocks may be received by HBEA (Figure 4, panel 3). Then, the rapid rise in total demand connectedness from August to September can be allocated to all the markets except CQEA (Figure 3 and Figure 4, panel 1), which does not contribute much influence but receives a lot. At the same time, the components of the variance variation of SZEA (Figure 4, panel 2) and HBEA (Figure 4, panel 3) due to the external shocks are relatively high. After September, as many markets have completed or are close to finishing the compliance matters, the degree of interaction between demand begins to decrease.

### 4.2. Connectedness in Frequency Domain

Frequency connectedness is also an important aspect for the interactions among China’s emission-trading markets. On the one hand, the fluctuations in demand for emission allowances sometimes tend to be relatively more persistent, which can be seen from Figure 1, so whether the impact of such volatility caused by exogenous shocks in other markets is permanent or transient is a question that needs to be paid attention to. On the other hand, among the interactions with different continuity caused by the cross-market transmissions of exogenous shocks, the one that is more prominent and dominates the linkage of the entire system also needs to be focused on. This is convenient for policy makers to understand the interactive relationship of the emission-trading market in terms of the time structure, so that they can manage the trading behavior of the market while understanding the tolerance of the entire system to shocks. Based on these considerations, the frequency decomposition of the connectedness is carried out to obtain its dynamics in different time domains. Here, the analysis is mainly focused on the total demand connectedness to clarify which period contributes most to the connectedness system, because the directional connectedness is also calculated based on the total connectedness, which share the same frequency characteristics.

The three time bands of short-term, medium-term and long-term are firstly considered to perform the frequency decomposition. The choice of the time frequency is mainly based on the following aspects. First of all, China’s emission trading is mainly conducted on weekdays, and 1–5 days of market information generally needs to undergo 2 rest days before continuing to spread in the market, so we set less than 5 days as the short-term frequency. Second, the state of the publicly traded market changes very rapidly, so shocks that are in play for more than half of a month can be regarded as relatively long-lasting information. We hence consider 10 days and above as a relatively long-term frequency. The middle 5–10 is considered the intermediate frequency. The results of the frequency connectedness are displayed in Figure 5. It can be seen that the connectedness of the short-term basically ranges around 5%, the values of the medium-term are all close to zero, while the connectedness of the long-term is much similar to the total one without frequency decompositions, no matter the value or the trend. Then, due to the medium component accounting for a small portion of the total connectedness, the time band of 10 days-infinite is maintained to express the long-term effect, and the other two are collapsed to re-express a shorter term relative to the longer one. Hence, another estimation result under this specification is collected in Figure 6, which shows a long-term component a bit closer to the total demand connectedness.

Although other types of time bands are also tried, such as allocating more days to the relatively shorter term, they are finally abandoned due to the consideration of trying to express the short-term by as few days as possible. Hence, empirically speaking, it seems that the long-term effect more than 10 days dominates the connectedness of the emission-trading system, which implies that most of the impacts on the demand uncertainty arising from elsewhere could last for 10 days or more. Additionally, a downward trend of connectedness in the band of more than 10 days basically corresponds to an upward trend of the short-term components, which are almost always associated with periods of inactivity in Figure 1, and illustrate that market interactions are relatively more sustained when all markets are active. When some markets are not active, it is difficult for the mutual influence between the markets to be transmitted continuously, so the effect in the short-term will be highlighted to a certain extent, especially in the period from mid-2018 to early 2020.

### 4.3. Sensitivity Analysis

The connectedness of the demand in China’s emission-trading system is estimated through an already proposed framework, and two key parameters during the model specification are chosen to ensure that the calculations can be carried out, which are, individually, horizons for error forecasting and lags for the VAR process. To perform a sensitivity analysis is to clarify whether the variation of these two parameters can severely change the estimation results of connectedness, which mainly follows the methods of Diebold and Yilmaz (2012, 2015) [36,37]. Hence, in order to determine from the sensitivity analysis whether the selected parameters in the connectedness estimation are acceptable, some simulations are performed with the variation range of these two parameters containing the values used in Section 4.1 and Section 4.2. So, the error forecast horizon here is allowed to range from 50 to 125 accompanied by VAR lags changing from 1 to 6. The simulation results are collected in Figure 7 and classified by the forecast horizons. The dark gray band in each panel represents the scale that the estimated total connectedness is shifting with the lags of VAR ranging from 1 to 6, and the black curve is the connectedness with the medium lags over the range of 1 to 6, which is also the actual value selected in Section 4.1 and Section 4.2. It can be seen that the connectedness is almost not sensitive to the choice of the error forecast horizon, but more lags selected in the VAR process can result in a higher level of connectedness. Additionally, the change in forecast horizons may vary the extreme value of connectedness with higher lags of VAR. However, in fact, the 4-lag VAR process we selected according to the information criterion is not sensitive to the forecast horizon.

## 5. Conclusions

The dual-track emission trading of regional pilots in parallel with national (power industry) markets is already operating steadily in China. However, at present, China’s carbon-emission-trading system cannot be regarded as fully mature due to the obvious differences in transaction activity among several markets. For the robust implementation of a future multi-sectoral expansion of national trading, it is necessary to clearly understand the characteristics of traders’ demand for allowances across the regional pilots. Therefore, the purpose of this study is to find the shocks or situations that can make the demand for emission allowances in these markets mutually react. In order to achieve what we presented here, this study applies the connectedness framework to measure the degree of demand interactions across the regional trading markets in the face of uncertain shocks, and analyzes the variation of connectedness to clarify what factors the carbon markets all respond to. Additionally, in order to find a time frequency that mainly dominates the interactions among the emission-trading markets, and further to clarify the persistency of the shocks faced by these markets, frequency decomposition is carried out. The main findings and contributions from the empirical analysis are as follows.

First of all, this study estimates the demand connectedness among regional emission-trading markets and obtains its dynamic trend. The total demand connectedness ranges from 10% to 24% with the average value being close to 20%, which is similar to the degree of price spillover obtained by Zhao et al., (2020), Guo and Feng (2021) and Xiao et al., (2022), and relatively lower compared with financial asset markets [26,29,30]. However, the degree of demand connectedness is relatively higher among them, which indicates that the volume captures more commonality of trading behavior among regional pilots. Secondly, the driving force of the time-varying characteristic of connectedness is revealed. Similar to Guo and Feng (2021), institutional events of emission accounting, emission verification and allowance compliance play cyclical roles in the variations of cross-market interactions of allowance demands [29]. What is more, the directional connectedness reveals the contributions made by each market to the entire system. The specific performance of each market is not exactly the same as the conclusions reached by Zhao et al., (2020), Zhu et al., (2020), and Zhang and Zhang (2020) [26,27,28], but the situation of certain markets is similar to Guo and Feng (2021) and Xiao et al., (2022), that is, the Tianjin market has relatively little interaction with others [29,30]. Finally, the frequency distributions of connectedness are clarified. Unlike the price connectedness obtained by Guo and Feng (2021) [29], the medium-term effect of 5–10 days is close to 0, and the long-term effect of more than 10 days accounts for the absolute majority of the proportion of frequency bands, which indicates that the demand interactions among the emission-trading system due to the transmission of non-own shocks may last at least 10 days.

## 6. Implications

The empirical results could also suggest some policy implications. For policy makers, relevant regulatory authorities should pay more attention to monitoring market transactions during the period when the market is experiencing accounting, verification and compliance matters. Participating companies in various markets are relatively more sensitive to these institutional events. Strengthened supervision is very helpful to avoid large market fluctuations caused by abnormal trading behavior. Moreover, such regulatory measures require the coordination of relevant government departments in various regions. Because whether it is in the current segmented pilots or the future national unified market, it cannot be accurately predicted in advance which regions’ traders will be more sensitive. Therefore, it is better for the authorities related to carbon trading in various places to foster cooperation or to establish a more unified supervision system. Finally, in the process of continuously improving the carbon-trading market, policy makers should consider relatively long-term construction goals and implement relatively long-term policy shocks in the market, because short-term shocks may not cause market reactions. Meanwhile, for emission traders, it would be better to enter the CO_2_ emission accounting work in advance and to prepare sufficient emission allowances before verification and compliance having an impact on market prices, so as to try to bear as little extreme uncertainty as possible. Additionally, the CO_2_ emitters involved in emission trading could try to develop their carbon-trading strategy and risk-management agency, taking advantage of the additional emission allowances generated by continuous technological advances to obtain excess benefits in the trading market.

While, since the regional emission-trading pilot only covers a small part of China, it does not perfectly meet the idea we want to consider for the whole country. And since the national (power industry) emission market has only been in operation for more than one year, the generated transaction data are not enough to support the establishment of an empirical analysis over a long time span. Therefore, the study can only draw partial conclusions and use them to potentially extrapolate the scenario of the national trade. Additionally, the study can only identify institutional events of carbon markets for the dynamic drivers of cross-market interactions. But in fact, there are many factors that can drive market conditions, such as extreme weather, economic shocks, etc. However, these factors are not sufficiently represented in these connectedness measures and are difficult to clearly identify, but these limitations also inspire a more holistic and rational approach to investigate the emission-trading policy and its shocks in the economy. For example, professional climate–energy–economic models can be employed to simulate the impact of different emission allowance supplies and their prices on the climate governance of the economy.

## Figures and Tables

**Figure 1 ijerph-19-14053-f001:**
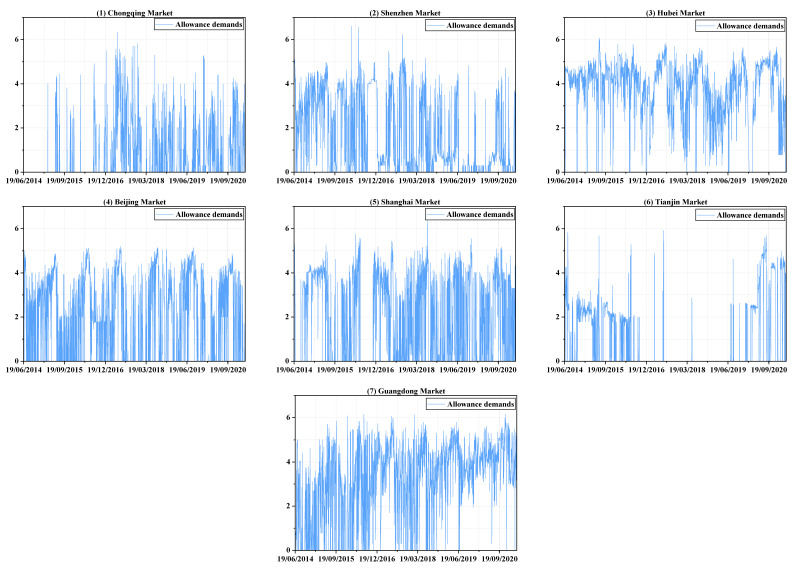
The dynamics of logarithmic trading volumes in seven markets.

**Figure 2 ijerph-19-14053-f002:**
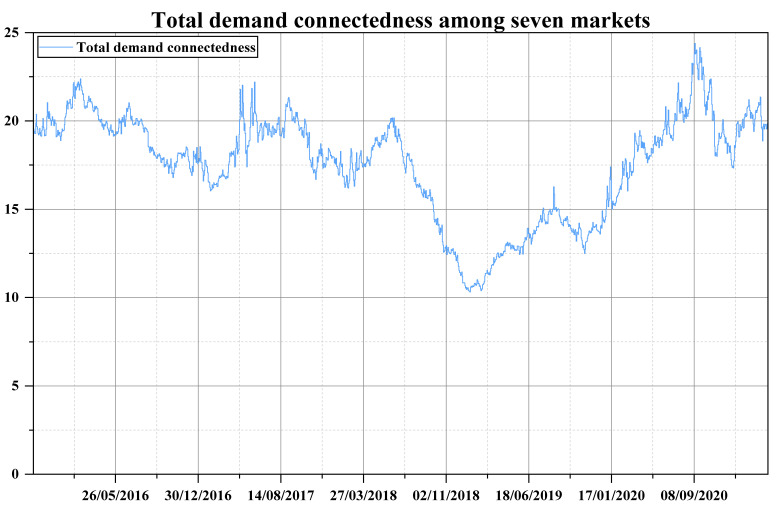
The dynamics of total demand connectedness.

**Figure 3 ijerph-19-14053-f003:**
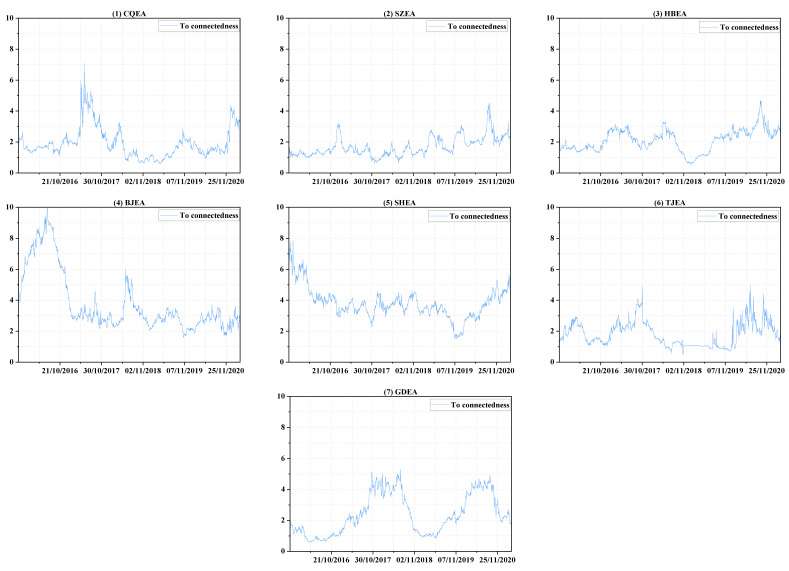
The dynamics of “to connectedness”.

**Figure 4 ijerph-19-14053-f004:**
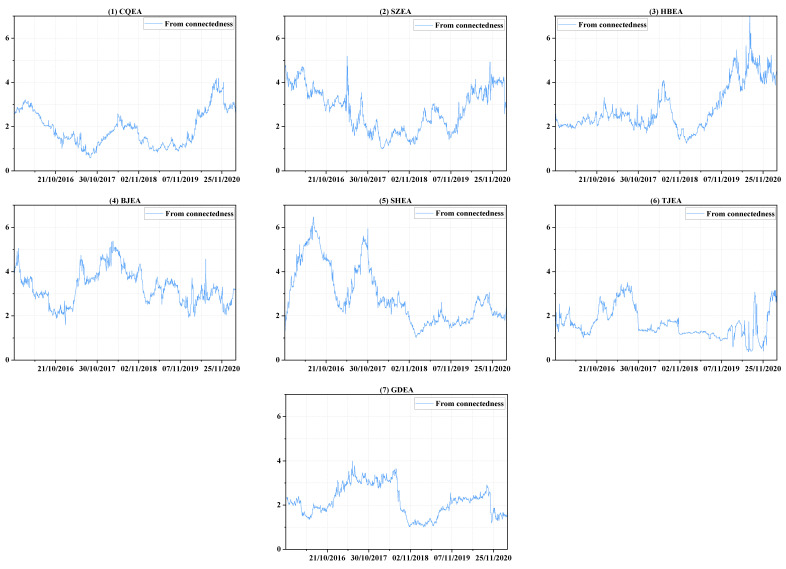
The dynamics of “from connectedness”.

**Figure 5 ijerph-19-14053-f005:**
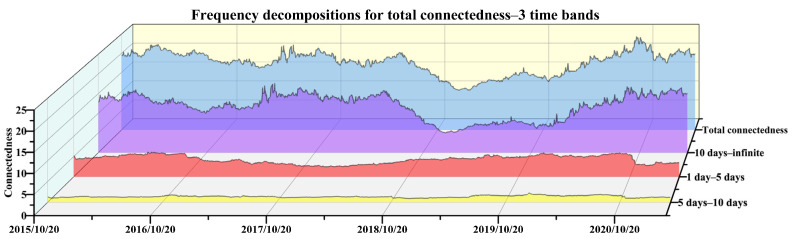
Total demand connectedness and frequency decomposition with three time bands.

**Figure 6 ijerph-19-14053-f006:**
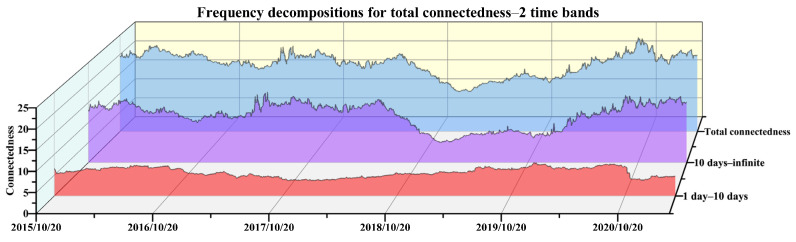
Total demand connectedness and frequency decomposition with two time bands.

**Figure 7 ijerph-19-14053-f007:**
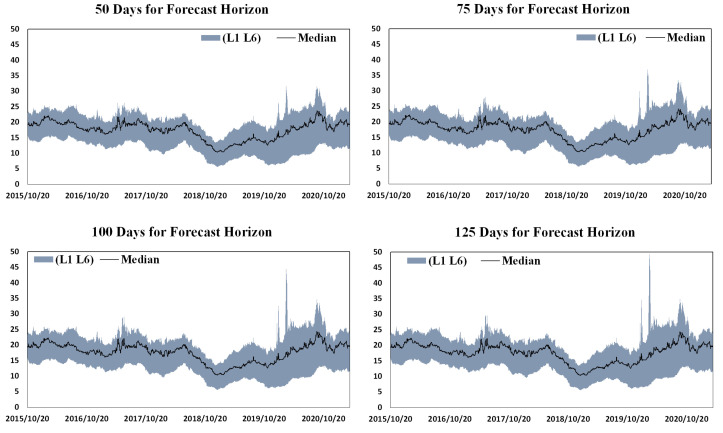
Total demand connectedness, error forecast horizons and VAR lags. Note: L1 and L6 individually represent the 1 and 6 lags for VAR system, and the median represents the medium lags during the range of 1 to 6, which is also the actual value selected in Section 4.1 and Section 4.2.

**Table 1 ijerph-19-14053-t001:** Descriptive statistics and previous test of the demand in seven markets.

Obs = 1663	Mean	Max	Min	Std. Dev.	ADF(C)	ADF(T)	PP(C)	PP(T)
CQEA	0.79	6.32	0	1.32	0.00 ***	0.00 ***	0.00 ***	0.00 ***
SZEA	1.75	6.60	0	1.75	0.00 ***	0.00 ***	0.00 ***	0.00 ***
HBEA	3.81	6.07	0	1.29	0.00 ***	0.00 ***	0.00 ***	0.00 ***
BJEA	2.02	5.19	0	1.84	0.00 ***	0.00 ***	0.00 ***	0.00 ***
SHEA	1.93	6.37	0	1.91	0.00 ***	0.00 ***	0.00 ***	0.00 ***
TJEA	0.97	5.92	0	1.50	0.00 ***	0.00 ***	0.00 ***	0.00 ***
GDEA	3.24	6.15	0	1.73	0.00 ***	0.00 ***	0.00 ***	0.00 ***

Values of ADF and PP individually represent the *p*-value of Augmented Dickey–Fuller and Phillips–Perron unit root test, (C) means the test equations contain a constant, and (T) means the test equations contain both a constant and time trend. *** indicates the rejection of null hypothesis of non-stationarity at the 1% significance level.

## Data Availability

The datasets used in this study are available from the corresponding author on reasonable request.

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
