# Peer review of "Measuring the Demand Connectedness among China’s Regional Carbon Markets"

_ijerph, 2022, doi:10.3390/ijerph192114053_

Round 1
Reviewer 1 Report
This paper analyses whether the transaction behaviors in various China’s carbon pilots have certain commonalities, the empirical results reveal the existence and its duration of such characteristic among China’s regional emission trading pilots, the institutional events like emission accounting, emission verification and allowance compliance mainly drive the time-varying volatility of the such cross-pilot interactions.
Overall, the paper is in a well-written structure, and I recommend a minor revision according to my comments/suggestions, which may be of use to the author(s):
(1) Introduction
It would be good to elaborate more the contributions of this paper in the introduction section.
(2) Literature review
The most recent research referring to this topic has been updated, please supplement the literature review section with them.
(3) Page 6, line 183-188
The presentation of the sample data should specify the reasons for the selection of the seven pilots (for example, how is China's current emissions trading system structured, and why some parts of it are not included in the sample?).
(4) Page 12, Figure7
Some annotations should be added to the figure to explain what "L1", "L6" and "Median" represent respectively, after all, the simple text in the legend has limited explanatory power.
(5) Section 4.2
In this section, it would be better to briefly explain why these three time bands, 1 day to 5 days, 5 days to 10 days, and 10 days to infinite, were defined.
(6) The results need to be compared with studies corresponding to similar topics, please confirm the similarities and differences between the results and that of literature, and highlight whether these results offer a new contribution to previous studies.
(7) The paper needs linguistic correction and improvement. In addition, there are some typos in the text (for example, "4-lagr" in page 12 line 405).
Author Response
Dear Reviewer,
Thank you for your time spent in reading and reviewing our paper and providing these comments for us. We have made every effort to address these issues. Please see the new version of the manuscript and read our point-by-point responses for details in the “Response document”.
Thank you for your work.
Best wishes,
Yours sincerely,
Chao Feng, Ph.D.

Reviewer 2 Report
The topic of demand connectedness among China's regional carbon markets is quite interesting, and the study topic is also very appropriate for the journal's direction. This work, in my viewpoint, has not yet satisfied the conditions for publication. The following are some of my alteration ideas for your consideration only.
Abstract: In this section, the authors may need to briefly introduce the sources and scope of the research data.
Introduction: Clearly, the author lacks a broad description of the available studies in this section. The article proposes research questions based on an investigation of the study background and pertinent data (Figure 1), however, there is no examination of topic-related theories and literature. A solid research question should be presented from the standpoints of theory and practice. I'm hoping the author can elaborate on this point.
Literature review: This section's writing logic must be clarified. The writers may, for example, consider 2.1 Demand connectedness among carbon markets; 2.2 Connectedness framework; and 2.3 Review summary.
Conclusion and implications: This section's writing logic needs to be enhanced. First and foremost, the author fails to address the paper's theoretical contribution in this section. The authors should compare their findings to previous research. Then, discuss their research contributions (often valuable findings that are different from those of others). In fact, reiterating the results in this second paragraph (section 5) is unnecessary. Second, the authors propose policy implications only based on the study's findings. Some stakeholders, such as companies involved in carbon emissions, may also need to assess if this research has practical implications, in my view. Finally, this study lacks self-criticism. There are evident research limitations in this study. The author should highlight these in the manuscript and provide feasible improvement strategies or future research directions.
Author Response

(The authors gave the same response as above.)

Reviewer 3 Report
The paper presents an analysis of the connectedness of China's regional carbon markets to support the sectoral expansion of the national market that is currently taking place based on those regional examples. The methodology is a composite VAR at various frequencies, which turns out to be a simplified version of currently existing bidimensional analyses. However, the frequencies selected are acceptable and seem representative. The article presents some flaws, mainly in form, that should be corrected. In this sense, I propose to accept the paper with minor corrections.
Observations to the paper:
There is already a China National ETS in place since 2021, only including the electricity sector. Assuming that, in the abstract, the authors refer to future multi-sectoral expansion, the concepts should be used with precision throughout the text. Thus, the paper measures the interconnection of regional markets, partly via the national market, which covers over 2000 companies in the electricity sector at the moment.
Given this national evolution of China's markets, it is also important to report, right from the start, the data used, which is of transacted volumes in regional markets between 2014 and 2021. The article does not consider transacted volumes in the national market, which also begins in 2021. This should be mentioned in the abstract, and, across the text, the analysis should consider the timings of China's domestic policies.
Regardless of the necessary corrections in the framework and interpretation of results, the paper presents a statistical analysis of variance decomposition relevant to understanding how regional markets connect and whether a future multi-sector national market will be functional. The contributions of the article, which result from the statistical analysis, are well identified in the article (57-60 pp2). This is the article's added value; hence, it is relevant to focus the literature review on the mentioned aspects. However, section 2 is mainly linked to the analysis of carbon market impacts, and only the last paragraph (118-128 pp3) refers to similar methodologies of this analysis and some studies of its application. This structure could be much improved, essentially developing the methodologies of frequency measurement and their applications, namely time-frequency analysis studies, including wavelets, in the carbon price analysis. From here, the authors could justify the choice of the methodology more robustly, even because they present a way of doing manually what some more complete methodologies already do, as in the case of wavelets.
As for the VAR methodology, as indicated by the authors, it is pretty standardized in econometric analyses, including in analyses of global carbon markets. Thus, it does not seem relevant to present the detail of the analysis on pages 4-6. A relevant aspect of the VAR analysis concerns the order of impacts, considered via the Cholesky decomposition order, which is not mentioned in the article. Sometimes a change in the order of impacts can affect the final results, which should be recognized, and eventually analyzed.
Finally, the conclusions are concrete and valuable for constructing China's national carbon market.
Generally, English is reasonable, but the type of language could be improved at some points. E.g. lines 51-53, page 2.
Author Response

(The authors gave the same response as above.)

Round 2
Reviewer 2 Report
This version is much better. It is clear that the authors have gone through a thorough revision process. However, it is apparent that this paper's theoretical contribution might be developed further. But even so, at this point, this manuscript is still quite near to publishing quality.